Road surface semantic segmentation for autonomous driving

Zhao Huaqi 1
Wang Su 1
Peng Xiang 1
Pan Jeng-Shyang 2
Wang Rui 3
Liu Xiaomin 1 xiaominliu@vip.sina.com
1 The Heilongjiang Provincial Key Laboratory of Autonomous Intelligence and Information Processing, School of Information and Electronic Technology, Jiamusi University , Jiamusi, Heilongjiang , China
2 College of Computer Science and Engineering, Shandong University of Science and Technology , Qingdao, Shandong , China
3 Dongfeng District People’s Court , Jiamusi, Heilongjiang , China
Coelho Paulo Jorge
Electronic publication date: 2024 Sep 25
Publication date: 2024
Volume: 10
Electronic Location ID: e2250
Received 2024 May 28; Accepted 2024 Jul 19
Copyright: © 2024 Zhao et al.
Copyright year: 2024
Copyright holder: Zhao et al.
License: This is an open access article distributed under the terms of the Creative Commons Attribution License, which permits unrestricted use, distribution, reproduction and adaptation in any medium and for any purpose provided that it is properly attributed. For attribution, the original author(s), title, publication source (PeerJ Computer Science) and either DOI or URL of the article must be cited.
License URL: https://creativecommons.org/licenses/by/4.0/

Keywords: Semantic segmentation, Transformer, Weight-sharing factorized attention, Cross-attention combining spatial and frequency features, Parallel-gated feedforward network

Funding: National Natural Science Foundation of China 51278227 Natural Science Foundation of Heilongjiang LH2022F052 National Natural Science Foundation Training Project of Jiamusi University JMSUGPZR2022-015 Space-Land Collaborative Smart Agriculture Innovation Team 2023-KYYWF-0638 Jiamusi University DJXSTD202417 Doctoral Program of Jiamusi University JMSUBZ2022-13 This work was supported by the National Natural Science Foundation of China project (51278227), the Natural Science Foundation of Heilongjiang (LH2022F052), the National Natural Science Foundation Training Project of Jiamusi University (JMSUGPZR2022-015), the Space-Land Collaborative Smart Agriculture Innovation team (2023-KYYWF-0638), the Jiamusi University "East Pole" academic team project (DJXSTD202417) and the Doctoral Program of Jiamusi University (JMSUBZ2022-13). The funders had no role in study design, data collection and analysis, decision to publish, or preparation of the manuscript.

==============================
Although semantic segmentation is widely employed in autonomous driving, its performance in segmenting road surfaces falls short in complex traffic environments. This study proposes a frequency-based semantic segmentation with a transformer (FSSFormer) based on the sensitivity of semantic segmentation to frequency information. Specifically, we propose a weight-sharing factorized attention to select important frequency features that can improve the segmentation performance of overlapping targets. Moreover, to address boundary information loss, we used a cross-attention method combining spatial and frequency features to obtain further detailed pixel information. To improve the segmentation accuracy in complex road scenarios, we adopted a parallel-gated feedforward network segmentation method to encode the position information. Extensive experiments demonstrate that the mIoU of FSSFormer increased by 2% compared with existing segmentation methods on the Cityscapes dataset.

Introduction

With advances in self-driving technology, stable road scene segmentation is crucial for the safe operation of autonomous driving systems. Semantic segmentation provides technical support for road surface segmentation tasks. It is used to classify all pixel labels of an image and has two characteristics compared to other vision tasks: pixel-by-pixel dense prediction and multi-class representation (Dong, Wang & Wang, 2023). Although semantic segmentation methods have achieved some good results in general pixel classification tasks, they perform poorly in road surface segmentation tasks for complex urban scenes. The reason is that these methods cannot mine pixel details and long-range context information of an image (Duong, Nguyen & Jeon, 2021). Figure 1 shows one of the complex road scenes which include complex intersections (large number of pedestrians and vehicles), variable road conditions in bad weather, etc. Therefore, enhancing the performance of road surface segmentation for complex scenes remains challenging.

Figure 1 An example of a complex urban scene.

This image is taken by our team.

Traditional road surface segmentation methods utilize manually extracted features to solve pixel-level label assignment problems,such as threshold selection (Otsu, 1979), superpixel algorithms (Achanta et al., 2012), and graph algorithms (Boykov & Jolly, 2001). With the development of deep learning, various methods based on fully convolutional networks (FCN) perform well in semantic segmentation tasks (Deng et al., 2022). DeepLabV3+ and PSPNet expand the receptive field by introducing a pooling module based on a spatial pyramid to integrate the features at different levels (Chen et al., 2017; Zhao et al., 2017). An HRNet can enhance semantic information by combining multiple high-resolution branches for feature interaction (Wang et al., 2020b). OCNet enhances the feature output of a backbone network through a global query context (Yuan, Chen & Wang, 2020). However, semantic segmentation methods based solely on the use of convolutions cannot establish effective context dependence on remote pixels in the image. Therefore, segmentation performance is degraded in complex and messy road scenes.

Recently, transformers have shown promising performances in semantic segmentation (Dosovitskiy et al., 2020; Liu et al., 2021; Touvron et al., 2021; Wang et al., 2021, 2022b). DPT improves the performance of dense prediction tasks by building transformer-based encoders (Ranftl, Bochkovskiy & Koltun, 2021). SETR introduces a sequence-to-sequence approach that utilizes a pre-trained vision transformer (Vit) to extract features (Zheng et al., 2021). However, SETR does not downsample the spatial resolution, which requires considerable computation. SegFormer enhances efficiency by incorporating an encoder based on a hierarchical transformer and a lightweight decoder (Xie et al., 2021). A series of semantic segmentation methods with transformers uses self-attention to update the semantic information of an image. However, self-attention has a high computational cost, which makes it unsuitable for realistic scenarios (Dosovitskiy et al., 2020).

However, it is very difficult to simplify the complexity of transformer from the perspective of spatial domain. Inspired by the fact that frequency features perform well in classification tasks (Rao et al., 2021; Wang et al., 2020a), we find that semantic segmentation is also very sensitive to frequency features. Thus, we propose an important frequency feature extraction method to directly capture the frequency features in the spatial domain by constructing a dynamic frequency capture module.

The existing road scene semantic segmentation methods have the following shortcomings: 1. These methods cannot establish context dependence on the remote pixels of an image, resulting in low segmentation performance for overlapping or incomplete objects in the road scene (Cira et al., 2022; Tian et al., 2022).

2. Existing road surface segmentation methods only focus on the spatial features of an image and do not consider the feature interactions between different domains, which results in boundary information loss for the segmented object (Vachmanus et al., 2020; Tian et al., 2022).

3. The methods cannot encode location information, resulting in poor target segmentation performance in complex road scenes (Cira et al., 2022; Vachmanus et al., 2020).

To solve these problems, we propose a frequency-based semantic segmentation with a transformer (FSSFormer) and the experiment is carried out around three parts parameter analysis, ablation experiments and comparative experiments. Also, compared with other segmentation methods, FSSFormer has a significant improvement in the evaluation metrics mIoU and FPS on four publicly available datasets. Besides, FSSFormer makes three main contributions. 1. Weight-sharing factorized attention (WSFA) is proposed to select important frequency features. This method constructs a dynamic frequency-capture module that enhances the differences between categories to enhance the segmentation accuracy of overlapping objects.

2. A cross-attention method combining spatial and frequency features is proposed to further extract detailed pixel information. This method obtains the boundary information of a segmented object by realizing the feature interactions between the spatial and frequency domains.

3. A parallel-gated feedforward network segmentation method is proposed to encode the location information. This method improves the segmentation performance of a target in complex road scenarios by learning the local structures of images.

Related work

Frequency feature extraction methods

Recently, scholars have found that high-level contextual semantic information contained in the frequency domain can help semantic segmentation methods learn the differences between categories, which makes the segmentation boundaries between different objects clearer (Dong, Wang & Wang, 2023). The WDSBLN obtains the deep features of SAR images to achieve better classification performance by analyzing the frequency information (Ni et al., 2023). Rao et al. (2021) proposed a global filter network to capture frequency features and obtain better image classification results. Dong, Wang & Wang (2023) proposed an adaptive frequency filter to extract frequency features that preserve contextual semantic information in high-resolution features. Li et al. (2021) proposed a discriminant feature learning framework for frequency perception to mine the frequency information of images. Various frequency feature extraction methods obtain high-level semantic information from images in the frequency domain. Therefore, they have important application value for semantic segmentation to extract frequency features.

Cross-attention

Scholars have proposed cross-attention by realizing the interaction of information between different branches or different modules (Wang et al., 2022a). For example, Wang et al. (2022a) adopted cross-attention between different resolutions to fully realize the interaction of the semantic information of low- and high-resolution branches. Chen, Fan & Panda (2021) proposed a token fusion strategy with cross-attention to extract multiscale features. Wei et al. (2020) extracted the correlation features within and between modalities using cross-attention. Zhu et al. (2022) designed a dual cross-attention method for learning subtle features and identifying fine-grained targets. Lin et al. (2022) achieved good performance with low computational cost by building a hierarchical network of a cross-attention transformer (CAT). However, these cross-attention methods realize the interaction of information in the spatial domain, which results in a limited receptive field for features. Therefore, the use of cross-attention between different domains is of great research significance.

Feedforward network

As a component of the transformer, a typical feedforward network cannot output high-quality features owing to its simple structure, resulting in poor generalization performance of segmentation methods (Zamir et al., 2022). Zamir et al. (2022) designed a gated feedforward network based on depthwise convolution to perform feature conversion. Xie et al. (2021) introduced a 3×3 depth-wise convolution in a feedforward network to provide location information. Dauphin et al. (2017) used a simplified gating mechanism in feedforward networks to capture the local contextual relationships between features. Feed-forward networks with a single gating mechanism cannot yield powerful representations, leading to poor segmentation performance. Therefore, we propose a parallel-gated feedforward network segmentation method that improves the segmentation performance of a target in complex road scenarios by learning the local structures of images.

Frequency-based semantic segmentation with transformer

Existing road-surface segmentation methods cannot capture complete contextual semantic information, leading to a decline in the segmentation performance of overlapping objects in road scenes. Moreover, these methods ignore the combination of spatial and frequency features and lose considerable edge-detail information. Furthermore, existing segmentation methods cannot encode location information, which decreases the segmentation performance of complex road surfaces. Therefore, we propose frequency-based semantic segmentation with Transformer (FSSFormer). The framework of the FSSFormer is shown in Fig. 2. We start with an input image and apply a convolutional layer and three successive residual layers (He et al., 2021) to obtain the reduced-resolution feature X^, and X^ is input to the dynamic frequency capture module to generate the frequency features X^A; X^ is then input into the linear attention operator to convert X^ into the spatial feature X^s, and next the frequency feature X^A and X^s are input into the cross-attention combining spatial and frequency features module to generate the mixed feature X^F; finally, X^F is input into the parallel-gated feedforward network module to generate the deep feature X^P, and X^P is input into the segmentation head to output the segmented image. Based on the above content, we study three parts: the important frequency feature extraction method based on weight-sharing factorized attention, the cross-attention method combining spatial and frequency features, and the parallel-gated feedforward network segmentation method.

Figure 2 The architecture of proposed segmentation method.

Important frequency feature extraction method based on weight-sharing factorized attention

Road surface segmentation is a highly complex task in pixel-level classification that results in category confusion, leading to low segmentation performance of overlapping or incomplete objects in road scenes (Dong, Wang & Wang, 2023). Inspired by AFFormer, we propose an important frequency feature extraction method based on weight-sharing factorized attention (WSFA). By constructing a dynamic frequency capture module, the frequency features of the image are directly captured in the spatial domain, and WSFA is then used to select important frequency features dynamically.

The core components of the dynamic frequency capture module are shown in Fig. 3, including the adaptive low-frequency capture kernel (ALFCK), adaptive high-frequency capture kernel (AHFCK), and WSFA. First, the feature X^ is generated by a convolutional layer and three consecutive residual layers, and then X^ is converted into X by dimensionality reduction, and X applies the ALFCK to obtain the low-frequency feature. Subsequently, X and the low-frequency feature are then converted into high-frequency features by the AHFCK. Finally, the low- and high-frequency features are aggregated to obtain frequency features, which are then applied to WSFA to generate the important frequency feature X^A. In the following section, ALFCK, AHFCK, and WSFA are introduced.

Figure 3 The structure of the dynamic frequency capture module.

Adaptive low-frequency capture kernel

Low-frequency features contain the most contextual semantic information in an image. In this study, average pooling was used as an adaptive low-frequency capture kernel to capture low-frequency features dynamically. Since different images have different cut-off frequencies, “adaptive” means that different groups of pooling are set to capture the low-frequency features according to the kernel size and step size. Given the input X^∈RB×C4×H16×W16. The formula for the adaptive low-frequency capture kernel is as follows:

(1) X=reshape(X^)

(2) ALF(X)=B(concat(φs×s(xm))).

In Eq. (1), reshape(⋅) represents dimension conversion; X∈RB×(H16∗W16)×C4. In Eq. (2), xm represents dividing the given feature into m groups; φs×s(⋅) represents an adaptive average pooling with an output size of S×S; concat(⋅) represents splicing; and B(⋅) represents the upsampling operation of bilinear interpolation.

Adaptive high-frequency capture kernel

The high-frequency features of images are key to retaining their details during semantic segmentation. To reduce computational complexity, this study directly utilizes low-frequency features to dynamically capture the high-frequency features of different images, which are expressed as

(3) AHF(X)=X∗(X−ALF(X))

where X represents the features projected onto lower dimensions. ALF(X) represents low-frequency features. This method obtains high-frequency features by subtracting low-frequency features from the original image features (Jiang, 2018). Moreover, segmentation noise is suppressed by using the Hadamard product of the original and high-frequency features (Dong, Wang & Wang, 2023).

Weight-sharing factorized attention

For high-frequency and low-frequency features, our goal is to select the key frequency features that can capture global contextual semantic information. Therefore, this study proposes WSFA. By designing an external, learnable, and shared weight space RS, the correlation between all the frequency features of the image is implicitly considered to select the important frequency features that are helpful for semantic segmentation.

Factorized attention uses the identity and Softmax(⋅) functions to factorize the Softmax attention map of self-attention approximately. Factorized attention first computes the matrix multiplication of the key vector K and value vector V. Subsequently, the Softmax(⋅) function is used to activate the matrix product, and finally, the matrix multiplication of the query vector Q and the result of the previous step are calculated. Given the input low-frequency feature ALF(X) and high-frequency feature AHF(X), the formulas for the factorized attention FA(Q,K,V) are defined as follows:

(4) XLH=concat(ALF(X),AHF(X))

(5) Q,K,V=Linear(WdXLH)

(6) FA(Q,K,V)=QC(Softmax(KT)∗V.

In Eq. (4), XLH represents the aggregated frequency features, and concat(⋅) represents the concatenation operation. In Eq. (5), Linear(⋅) is the learnable linear layer, and Wd represents 3×3 depth-wise convolution. In Eq. (6), C is the channel dimension of query vector Q. Moreover, factorized attention significantly degrades computational complexity by factorizing (Xu et al., 2021).

WSFA introduces an external weight RS based on the factorized attention. First, the external weight RS and the value vector V are matrix multiplied to obtain the attention map, and then the feature FA(Q,K,V), which are generated by the factorized attention, are multiplied with the attention map to generate the important frequency features. The formulas for WSFA EFA(Q,K,V) are as follows:

(7) EFA(Q,K,V)=FA(Q,K,V)∗Norm(V∗RS)

(8) X^A=reshape(EFA(Q,K,V))

where Norm(⋅) represents normalization, and RS∈RB×(H16∗W16)×C represents learnable weights. WSFA can select important frequency features that help capture high-level contextual semantic information by introducing a shared weight, RS.

In summary, we proposed an important frequency feature extraction method based on WSFA. In the dynamic frequency capture module, an adaptive low-frequency capture kernel and an adaptive high-frequency capture kernel are used to directly capture the low- and high-frequency features in the spatial domain, and WSFA is proposed to select and enhance important frequency features.

Cross-attention method combining spatial and frequency features

Existing semantic segmentation methods extract only spatial features and ignore frequency features, resulting in the loss of detailed image information. Therefore, we propose a cross-attention method combining spatial and frequency features, which realizes the interaction of spatial and frequency features to obtain segmentation edge detail information. The framework of cross-attention, which combines spatial and frequency features, is shown in Fig. 4. The inputs are the spatial feature XS and frequency feature X^A, and the cross-attention mechanism is then applied to generate the mixed feature X^F of different domains. The generation of frequency features is introduced. In the following section, we introduce the generation of spatial features using the linear attention operator module and the interaction of features between different domains with cross-attention combining spatial and frequency features.

Figure 4 The structure of the cross-attention combining spatial and frequency features.

The spatial feature XS∈RB×C4×H16×W16 is generated by the linear attention operator module (LAO) inspired by external attention (Guo et al., 2022). As shown in Fig. 5, the input of LAO is the reduced-resolution feature X^, and the output is the spatial feature XS. XS is defined as:

Figure 5 The structure of the linear attention operator.

(9) XS=DN(X^⋅KeT)⋅Ve

where Ke,Ve∈RM×D are learnable weight parameters, M is the resolution size of the feature, and DN(⋅) is the double normalization operation. Moreover, we eliminated the multihead mechanism of external attention to reduce the computational cost.

In the cross-attention combined spatial and frequency feature modules, the inputs were the spatial feature XS and frequency feature X^A. First, the input X^A applies a dimension transformation to generate XA∈R1×(B∗C4)×H16×W16. XA is expressed as:

(10) XA=reshape(X^A).

Second, to make the spatial features and frequency features better fusion in each dimension, the spatial feature X undergoes a series of matrix operations such as normalization, pooling, convolution, splitting along the channel dimension, and dimension conversion to generate two cross-feature vectors: KS∈R(B∗144)×C5 and VS∈R(B∗C5)×144. KS and VS are defined as follows:

(11) KS,VS=σ(θ(W1×1⋅Pooling(Norm(XS))))

where σ(⋅) and θ(⋅) represent dimension conversion and matrix splitting respectively, W1×1 represents 1×1 convolution, Pooling(⋅) represents the pooling operation, and Norm(⋅) represents the normalization.

Finally, since simple feature aggregation operations cannot realize feature interaction between different domains, a cross-attention operation is applied to generate the hybrid feature XF∈R1×(B∗C5)×H16×W16, and then XF performs dimension transformation to obtain the output X^F∈RB×C5×H16×W16. X^F is expressed as:

(12) XF=Softmax(XA⋅KSTdf)⋅VS

(13) X^F=reshape(XF)

where df represents the channel dimensions of XA. Moreover, when the space size of the cross feature is 12×12, our FSSFormer exhibits the best segmentation performance. The specific experimental details are presented in “Experiments”.

In summary, the cross-attention method that combines spatial and frequency features can obtain detailed image information through the interaction of the spatial and frequency features.

Parallel-gated feedforward network segmentation method

Existing segmentation methods perform well for simple scenes. However, owing to the poor generalization performance, the segmentation accuracy of complex road surfaces decreases. As the core component of a transformer, a feedforward network is typically composed of two fully connected layers and nonlinear activation functions. However, this network structure could only process pixels at different positions in the same manner (Xie et al., 2021). Because the pixel information at different positions is different, this structure cannot obtain the local information of images, resulting in poor generalization ability. Therefore, this article proposes a parallel-gated feedforward network segmentation method that improves the feature information flow in the feedforward network from two aspects: the parallel mechanism and the gated mechanism based on the GeLu activation function. The architecture of the parallel-gated feedforward network is shown in Fig. 6. First, the input mixed feature X^F is divided into two sets of features: XF1, XF2 by applying a parallel mechanism, and then XF1, XF2 are used as two parallel branches to generate features Y1, Y2, respectively, by applying a gated mechanism. Finally, Y1 and Y2 are aggregated to generate the enhanced feature X^P. Specific details of the parallel and gated mechanisms are presented below.

Figure 6 The structure of the parallel-gated feedforward network module.

Parallel mechanism refers to parallel computing and has two advantages. The parallel mechanism retains the advantages of the multihead mechanism in the transformer to a certain extent. Conversely, the relationship between pixels at different positions is captured by generating two different paths for feature mapping. Given an input feature vector X^F∈RB×C5×H16×W16, the formula is as follows:

(14) XF1,XF2=Split(X^F).

In Eq. (14), Split(⋅) represents splitting the feature into two parallel branch features; XF1 and XF2 represent the features of two parallel branches, respectively.

Moreover, to learn the local structure of images, a gated mechanism was designed as an element-wise product of two parallel branches, inspired by Restormer (Zamir et al., 2022). First, the two parallel branches use the BatchNorm normalization function and depthwise convolution to encode different pixel positions. Second, the GeLu activation function is used to activate the encoded features in the two parallel branches. Finally, the element product operation was applied to two parallel branches to realize the interaction of pixels at different positions. The formula used is as follows:

(15) Y1=ϕ(Wd1Wp1(BN(XF1)))⋅Wd2Wp2(BN(XF2))

(16) Y2=Wd1Wp1(BN(XF1))⋅ϕ(Wd2Wp2(BN(XF2)))

where Wp(⋅) represents 1×1 pixel convolution. Wd(⋅) represents 3×3 depth-wise convolution. ϕ(⋅) represents the GeLu activation function (Hendrycks & Gimpel, 2016). BN(⋅) represents batch normalization (Ioffe & Szegedy, 2015). The ⋅ represents element-wise multiplication.

Finally, the feature maps of the two parallel branches were spliced together, and the output feature was X^P, as shown in the following formula:

(17) EG(X)=concat(Y1,Y2)

(18) X^P=Wp0EG(XF)+XF

where concat(⋅) is the splicing operation.

The deep features X^P are then fed into the segmentation head, which consists of two convolutional layers, and the segmentation head is used to output the segmented image.

Generally, the proposed parallel-gated feedforward network segmentation method enhances the feature representation by encoding the position information and learning the local structure of the image, which improves the segmentation performance of the target in complex road scenes.

Experiments

We validated the proposed FSSFormer using four publicly available datasets: Cityscapes (Cordts et al., 2016), DarkZurich (Sakaridis, Dai & Van Gool, 2020), ACDC (Sakaridis, Dai & Van Gool, 2021) and COCO-Stuff (Caesar, Uijlings & Ferrari, 2018). This section focuses on three aspects: parameter analysis, ablation experiments, and comparative experiments. All experiments are conducted on a single RTX 2080Ti.

Experimental setup

We used the Paddle1.8.0 framework (Wang et al., 2022a) for the experiments and uniformly used the AdamW optimizer. The initial learning rate was set to 0.0004, and the weight attenuation was set to 0.0125. Additionally, we used Params, mIoU, precision, recall, and FPS to evaluate the segmentation performance. Params represents the number of model parameters, and mIoU represents the ratio of the intersection and union of two sets of true and predicted values. Precision is the probability that a given class is correct. Recall is the probability that a class is correctly predicted among the true values. The mIoU, precision, and recall were calculated as shown in Eqs. (19)–(21):

(19) mIoU=1k+1∑i=0kPii∑j=0kPij+∑j=0k(Pji−Pii)

(20) Precision=TP(TP+FP)

(21) Recall=TP(TP+FN).

In Eq. (19), the relationship between classes is defined as P, which is used to represent the probability of true and false positives of the pixels. In Eq. (20), TP represents true positives, and FP represents false positives. In Eq. (21), FN represents a false negative.

Moreover, FPS represents the number of pictures processed per second. The FPS is measured on a single RTX 2080Ti without tensorRT acceleration by default.

Parameter analysis

In this study, a weight-sharing factorized attention is designed to enhance the segmentation accuracy of overlapping targets. A cross-attention method combining spatial and frequency features is introduced to obtain boundary information. A parallel-gated feedforward network segmentation method is proposed to improve the segmentation performance of the target in complex scenes. This section presents a parameter analysis of the three modules.

Parameter analysis of the group number M of important frequency feature extraction method

To explore the impact of the group number M of the low-frequency capture kernel on the important frequency feature extraction method, this section uses group number M as a parameter for experimental analysis. The values of M range from 2 to 9 respectively.

As shown in Fig. 7, when the number of groups of low-frequency capture kernels was four, the mIoU was the highest, reaching 73.38 %, which was 3.25 % higher than the lowest. Within a certain range, the larger the value of M, the more low-frequency features of different frequency bands are captured. Therefore, the mIoU initially increased with an increase in M. Beyond a certain range, the captured low-frequency features contain more segmentation noise, which degrades the semantic segmentation performance. Therefore, as the value of M increased, the mIoU slowly decreased. Through the above experimental analysis, it is proven that the extraction of frequency features substantially improves segmentation performance.

Figure 7 Parameter analysis of the group number M of important frequency feature extraction method.

The red-dot data point shows that when the number of groups of low-frequency capture kernels was four, the mIoU was the highest, reaching 73.38%.

Parameter analysis of cross-feature space size of the cross-attention method combining spatial and frequency features

To explore the influence of different cross-feature space sizes on FSSFormer, this section sets the cross-feature space size S as the experimental parameter and designs eight groups of experiments with S=6×6, 8×8, 10×10, 12×12, 14×14, 16×16, 18×18 and 20×20. The experimental results are shown in Fig. 8.

Figure 8 Parameter analysis of cross-feature space size of cross-attention method combining spatial and frequency features.

The red-dot data point shows that when the size of the cross-feature space was 12 × 12, the mIoU reached the highest value of 73.38%.

As shown in Fig. 8, when the size of the cross-feature space was 12×12, the mIoU reached the highest value of 73.38 %, which was 0.82 % higher than the lowest value. Moreover, with an increase in the S value, the mIoU value also increases; when the S value is 144, it reaches a peak value of 73.38 %. With an increase in the S value, the mIoU value slowly decreases. When the space size of the spatial and frequency features are closer together, the features in different domains interact better. Because the space size of the feature in the frequency range was 12×12, the mIoU reached its maximum when S was set to 12×12. Based on the above experimental analysis, the cross-attention method combining spatial and frequency features can improve segmentation performance through the interaction of features between different domains.

Parameter analysis of depth-wise convolution of the parallel-gated feedforward network segmentation method

To explore the impact of different groups of depthwise convolutions in a parallel-gated feedforward network, group G was set as the experimental parameter, and eight groups of experiments were designed with G=8, 16, 32, 64, 128, 256, 512, and 1,024. The experimental results are shown in Fig. 9.

Figure 9 Parameter analysis of depth-wise convolutions of parallel-gated feedforward network segmentation method.

The red-dot data point shows that when G was 1,024, the mIoU value was 73.38%.

As shown in Fig. 9, the mIoU also increased with an increase in G. However, the larger the value of G, the higher the computational cost. When the value of G exceeded a certain range, the speed of the model significantly degraded. To balance speed and segmentation performance, 1,024 was set as the value of G. When G was 1,024, the mIoU value was 73.38 %, which was 1.74 % higher than the lowest value. This also proves that the use of different depthwise convolutions in a parallel-gated feedforward network impacts segmentation performance.

Ablation studies and analysis

Ablation experiments were conducted using the Cityscapes (Cordts et al., 2016) dataset. The training settings for the experiments described in this section are the same as those described above.

Ablation experiment results of each module

To prove the performance of the dynamic frequency capture module, the cross-attention combining spatial and frequency feature modules, and the parallel-gated feedforward network module proposed in this study, this section conducts experimental verification on the Cityscapes (Cordts et al., 2016) dataset.

In this section, the design and experiments for each module are described based on the transformer network. In Table 1, DFCM represents the dynamic frequency capture module, CACSF represents cross-attention combining spatial and frequency features, and PGFFN represents the parallel-gated feedforward network module. As shown in Table 1, DFCM improves the segmentation accuracy of overlapping objects by capturing high-level contextual semantic information; compared with the transformer-based segmentation method, mIoU is increased by 3.92 %. From the data analysis in Fig. 10, the design of the CACSF further improves the semantic segmentation performance by obtaining image boundary information, and the value of mIoU is 1.88 % higher than that without the CACSF. Finally, although the parallel-gated feedforward network module only improves the mIoU by 0.41 %, it can be seen from Fig. 10 that the precision of PGFFN reaches the highest, which indirectly proves that encoding the position information has substantial help in improving the semantic segmentation performance. Figure 10 illustrates the changes in recall and precision in the four sets of ablation experiments. Moreover, Experiments 3 and 4 had the highest recall and precision, respectively.

Table 1 The experimental results of proposed segmentation methods on Cityscapes.

	DFCM	CACSF	PGFFN	mIoU	
Experiment 1				67.17 %	
Experiment 2	✓			71.09 %	
Experiment 3	✓	✓		72.97 %	
Experiment 4	✓	✓	✓	73.38 %	

Figure 10 Changes in mIoU, precision and recall in ablation experiments.

Advantages of important frequency feature extraction method based on weight-sharing factorized attention

To prove that WSFA can capture high-level contextual semantic information and effectively segment the overlapping targets, this section presents experimental verification using different types of attention for a frequency feature extraction method based on WSFA.

As shown in Table 2, we found that WSFA was better than factorized attention in terms of speed and accuracy, and the mIoU value was improved by 3.96 %. Moreover, the efficiency of WSFA was much higher than that of self-attention, and the segmentation method with WSFA was almost twice as fast as that with self-attention. The experimental results show that WSFA can improve segmentation performance.

Table 2 The impact of our segmentation methods with different types of attention.

Method	GPU	FPS	mIoU	
Not using attention	RTX 2080Ti	74.2	63.09 %	
Factorized attention	RTX 2080Ti	73.1	69.42 %	
Self-attention	RTX 2080Ti	30.8	72.87 %	
Weight sharing factorized attention	RTX 2080Ti	73.7	73.38 %	

Advantages of cross-attention method combining spatial and frequency features

To verify the effectiveness of the cross-attention method, which combines spatial and frequency features, different cross-attention methods were used for experimental verification.

As shown in Table 3, the mIoU of the segmentation method with typical cross-attention is 1.03 % higher than that without cross-attention, indicating that the cross-attention mechanism is effective for the segmentation method. The mIoU of the segmentation method with cross-attention combining spatial and frequency features is 2.26 % higher than that with typical cross-attention, which indicates that cross-attention combining spatial and frequency features can obtain the boundary information of the segmented target, thereby improving the performance of semantic segmentation.

Table 3 The impact of our segmentation methods with different types of cross-attention.

Method	GPU	FPS	mIoU	
Not using cross-attention	RTX 2080Ti	74.5	68.09 %	
Typical cross-attention	RTX 2080Ti	74.1	69.12 %	
Cross-attention across space-frequency features	RTX 2080Ti	73.7	73.38 %	

Advantages of parallel-gated feedforward network segmentation method

To prove that the parallel-gated feedforward network segmentation method can enhance the segmentation accuracy of targets in complex scenes, we will study whether the gated or parallel mechanism is adopted.

As shown in Table 4, the mIoU of the segmentation method with the gated feedforward network was 0.46 % higher than that of the typical feedforward network, and the improvement in the segmentation performance was not significant. Moreover, the mIoU of the segmentation method with the parallel feedforward network was 1.58 % higher than that of the typical feedforward network, indicating that the parallel mechanism was effective for our segmentation method. Furthermore, the segmentation performance of our method with the parallel-gated feedforward network was significantly enhanced compared to that of the typical feedforward network, and the mIoU was increased by 3.58%. The experimental results show that the parallel-gated feedforward network can improve semantic segmentation performance by encoding location information.

Table 4 The impact of our segmentation method with different feedforward network.

Method	GPU	FPS	mIoU	
Typical feedforward network	RTX 2080Ti	73.9	69.8 %	
Gated feedforward network	RTX 2080Ti	73.8	70.26 %	
Parallel feedforward network	RTX 2080Ti	73.8	71.38 %	
Parallel gated feedforward network	RTX 2080Ti	73.7	73.38 %	

Comparison with state-of-the-art semantic segmentation methods

In this section, we compare FSSFormer with top-ranking semantic segmentation methods and conduct experiments on Cityscapes (Cordts et al., 2016), COCO-Stuff (Caesar, Uijlings & Ferrari, 2018), ACDC (Sakaridis, Dai & Van Gool, 2021) and DarkZurich datasets (Sakaridis, Dai & Van Gool, 2020).

Results on Cityscapes dataset

Previous works on semantic segmentation have used Cityscapes (Cordts et al., 2016) as a standard benchmark, considering its high-quality annotation. As shown in Table 5, we tested the speed of models published for nearly 2 years on our platform with the same settings for a fair comparison. The experimental results show that FSSFormer outperforms the current leaderboard SOTA methods (VLTSeg and PIDNet-L) in both speed and accuracy, increasing the accuracy from 72.5 % to 73.38 % mIOU, making it the most accurate model in the real-time domain. Also, transformer-based semantic segmentation methods, such as SegFormer, performed better than convolution-based semantic segmentation methods, such as DeepLabV3+ and PSPNet. However, the number of parameters in transformer-based semantic segmentation methods is large, making them unsuitable for real-world applications. Moreover, our FSSFormer achieved 73.38 % mIoU only with 7.8 M parameters. Compared with SegFormer, the number of parameters in FSSFormer was reduced by 5 M, and the mIoU was increased by 1 %. Furthermore, compared with lightweight semantic segmentation methods such as RTFormer, FSSFormer improves the mIoU by nearly 2 % while using only half of the model parameters. To reflect the training process of each method more intuitively, the changes in the mIoU values with an increase in model iterations are shown in Fig. 10.

Table 5 Comparison to semantic segmentation methods on Cityscapes.

Segmentation methods	GPU	Params	mIoU	Resolution	FPS	
BiSeNetV2 (Yu et al., 2018)	GTX 1080Ti	49.0 M	71.89 %	1,024×512	47.3	
SegFormer (Xie et al., 2021)	RTX 3090	84.7 M	72.38 %	1,024 × 512	48.6	
FCN (Long, Shelhamer & Darrell, 2015)	RTX 2080Ti	9.8 M	63.29 %	1,024×512	14.2	
OCRNet (Yuan, Chen & Wang, 2020)	RTX 2080Ti	10.5 M	67.7 %	1,024×512	30.3	
PSPNet (Zhao et al., 2017)	RTX 2080Ti	13.7 M	70.2 %	1,024×512	11.2	
DeepLab V3+ (Chen et al., 2018)	RTX 2080Ti	15.4 M	70.54 %	1,024×512	8.4	
RTFormer (Wang et al., 2022a)	RTX 2080Ti	16.8 M	71.13 %	1,024×512	71.4	
PIDNet-L (Xu, Xiong & Bhattacharyya, 2023)	RTX 2080Ti	10.3 M	72.13 %	1,536×768	73.2	
VLTSeg (Hümmer et al., 2023)	RTX 2080Ti	28.3 M	72.5 %	1,024×512	72.1	
DDRNet-23 (Hong et al., 2021)	RTX 2080Ti	20.1 M	72.6 %	1,024×512	75.2	
Ours	RTX 2080Ti	7.8 M	73.38 %	1,024×512	73.7	

As shown in Fig. 11, the overall curve of the proposed segmentation method is smoother than those of the other segmentation methods, indicating that our segmentation method is more stable throughout the training process. However, in the training process of the current leaderboard SOTA method PIDNet-L, there is a large fluctuation range, which indicates that the model is unstable. In the early training stage, the mIoU of the proposed segmentation method steadily increased. When the iteration is 80,000, the mIoU value of the proposed method is stable at approximately 71 %, and our segmentation method can complete the training process faster than the other segmentation methods. In summary, the segmentation performance of FSSFormer was better than that of the other semantic segmentation methods during training.

Figure 11 The mIoU changes of semantic segmentation methods at different iters on cityscapes.

Results on COCO-Stuff dataset

The COCO-Stuff dataset (Caesar, Uijlings & Ferrari, 2018) contains several intractable samples from the COCO dataset. For the COCO-Stuff dataset, only the Params of RTFormer, OCRNet, VLTSeg, PIDNet is comparable with our model, so we tested their speeds with the same settings on our platform for a fair comparison. As shown in Table 6, FSSFormer provides much higher accuracy compared with other models with similar inference speeds. FSSFormer outperforms the previous state-of-the-art model VLTSeg by 1.53 % mIOU with a speedup of about 0.1 ms per image. Also, compared with lightweight segmentation methods (such as RTFormer), the proposed segmentation method reduces the number of parameters by 10 M. In addition, compared with semantic segmentation methods based on transformer networks, such as SegFormer, the proposed segmentation method achieves 33.8 % mIoU with only 6.8 M Params, and the number of parameters is degraded by 10 times while the speed is nearly doubled. Compared with the current leaderboard SOTA method PIDNet-L, the number of parameters of FSSFormer was reduced by 3.5 M, and the mIoU was increased by 2.6 %. In summary, the proposed segmentation method achieved the best tradeoff between speed and segmentation performance.

Table 6 Comparison to semantic segmentation methods on COCO-Stuff.

Segmentation methods	GPU	Params	mIoU	Resolution	FPS	
BiSeNetV2 (Yu et al., 2018)	GTX 1080Ti	5.2 M	25.2 %	640×640	52.4	
SegFormer (Xie et al., 2021)	RTX 3090	84.7 M	36.7 %	640×640	50.3	
VLTSeg (Hümmer et al., 2023)	RTX 2080Ti	28.3 M	32.27 %	640×640	76.3	
DDRNet-23 (Hong et al., 2021)	RTX 2080Ti	20.1 M	32.1 %	640×640	74.3	
DeepLab V3+ (Chen et al., 2018)	RTX 2080Ti	17.4 M	31.54 %	640×640	12.5	
RTFormer (Wang et al., 2022a)	RTX 2080Ti	16.8 M	35.3 %	640×640	66.1	
PSPNet (Zhao et al., 2017)	RTX 2080Ti	13.7 M	30.2 %	640×640	21.3	
OCRNet (Yuan, Chen & Wang, 2020)	RTX 2080Ti	13.5 M	37.9 %	640×640	35.2	
PIDNet-L (Xu, Xiong & Bhattacharyya, 2023)	RTX 2080Ti	10.3 M	31.2 %	640×640	75.8	
FCN (Long, Shelhamer & Darrell, 2015)	RTX 2080Ti	9.8 M	28.71 %	640×640	19	
Ours	RTX 2080Ti	6.8 M	33.8 %	640×640	76.9	

As shown in Fig. 12, the trend of the proposed segmentation method steadily increased in the early training stage and flattened in the later stage. Although some semantic segmentation methods (such as SegFormer) have a higher mIoU than the proposed method, the change span of the mIoU is large in the later stages, which means that they are not stable throughout the training process. However, the two latest methods (VLTSeg and PIDNet-L) show a wide range of stagnation in mIoU during training. According to the analysis, the performance of FSSFormer on COCO-Stuff in terms of training stability was better than that of the other semantic segmentation methods.

Figure 12 The mIoU changes of semantic segmentation methods at different iters on COCO-Stuff.

Results on ACDC dataset

ACDC (Sakaridis, Dai & Van Gool, 2021) is a dataset of autonomous driving scenarios under adverse weather conditions. We conduct experiments on the ACDC dataset to test the generalization performance of our model. As shown in Table 7, our FSSFormer achieved 66.71 % mIoU only with 7.8 M parameters. Moreover, compared with SegFormer with the best segmentation performance, the inference speed per image of FSSFormer is increased by 10 ms, and the mIoU was increased by 0.88 %. Furthermore, compared with lightweight semantic segmentation methods such as FCN, FFSFormer improves the mIoU by nearly 3.49 % while using 7.8 M parameters. However, the current leaderboard SOTA method PIDNet-L, which performs well on Cityscapes (Cordts et al., 2016), has a mIoU of only 23.96 % on the ACDC dataset. The results show that our method is still quite advantageous under severe weather conditions.

Table 7 Comparison to semantic segmentation methods on ACDC.

Segmentation methods	GPU	Params	mIoU	Resolution	FPS	
BiSeNetV2 (Yu et al., 2018)	GTX 1080Ti	49.0 M	40.71 %	1,024×512	45.1	
SegFormer (Xie et al., 2021)	RTX 3090	12.8 M	65.83 %	1,536×768	45.3	
PIDNet-L (Xu, Xiong & Bhattacharyya, 2023)	RTX 2080Ti	10.3 M	23.96 %	1,536×768	72.4	
OCRNet (Yuan, Chen & Wang, 2020)	RTX 2080Ti	10.5 M	58.15 %	1,024×512	28.7	
DeepLab V3+ (Chen et al., 2018)	RTX 2080Ti	15.4 M	58.62 %	1,024×512	7.9	
PSPNet (Zhao et al., 2017)	RTX 2080Ti	13.7 M	59.31 %	1,024×512	10.6	
FCN (Long, Shelhamer & Darrell, 2015)	RTX 2080Ti	9.8 M	63.22 %	1,024×512	12.7	
DDRNet-23 (Hong et al., 2021)	RTX 2080Ti	20.1 M	63.59 %	1,024×512	71.5	
VLTSeg (Hümmer et al., 2023)	RTX 2080Ti	28.3 M	64.2 %	1,024×512	72.1	
RTFormer (Wang et al., 2022a)	RTX 2080Ti	16.8 M	64.29 %	1,024×512	69.2	
Ours	RTX 2080Ti	7.8 M	66.71 %	1,024×512	71.8	

Results on DarkZurich dataset

DarkZurich (Sakaridis, Dai & Van Gool, 2020) is a nighttime driving dataset for autonomous driving. Experiments are conducted on the DarkZurich dataset to test the performance of our model when driving at night. As shown in Table 8, compared with the VLTSeg with the best segmentation performance, the inference speed per image of FSSFormer is increased by 0.02 ms, and the mIoU was increased by 0.46 %. Besides, compared with the FCN, which has the least parameters, although our method has 2 M lower parameters, the mIoU was increased by 5.67 %. Moreover, our FSSFormer achieved 40.96 % mIoU only with 7.8 M parameters. Even in the dark environment, the segmentation performance of the proposed method is also higher than other methods. This reflects from the side that the generalization performance of our method is stronger than that of other methods.

Table 8 Comparison to semantic segmentation methods on DarkZurich.

Segmentation methods	GPU	Params	mIoU	Resolution	FPS	
BiSeNetV2 (Yu et al., 2018)	GTX 1080Ti	49.0 M	37.91 %	1,024×512	48.2	
SegFormer (Xie et al., 2021)	RTX 3090	12.8 M	40.38 %	1,536×768	46.7	
FCN (Long, Shelhamer & Darrell, 2015)	RTX 2080Ti	9.8 M	35.29 %	1,024×512	13.4	
DDRNet-23 (Hong et al., 2021)	RTX 2080Ti	20.1 M	36.6 %	1,024×512	71.5	
OCRNet (Yuan, Chen & Wang, 2020)	RTX 2080Ti	10.5 M	37.4 %	1,024×512	31.2	
RTFormer (Wang et al., 2022a)	RTX 2080Ti	16.8 M	37.52 %	1,024×512	72.6	
DeepLab V3+ (Chen et al., 2018)	RTX 2080Ti	15.4 M	38.26 %	1,024×512	9.3	
PSPNet (Zhao et al., 2017)	RTX 2080Ti	13.7 M	39.7 %	1,024×512	12.3	
PIDNet-L (Xu, Xiong & Bhattacharyya, 2023)	RTX 2080Ti	10.3 M	39.82 %	1,536×768	73.8	
VLTSeg (Hümmer et al., 2023)	RTX 2080Ti	28.3 M	40.5 %	1,024×512	73.2	
Ours	RTX 2080Ti	7.8 M	40.96 %	1,024×512	74.1	

Visualized result analysis

To verify the practical application ability of the proposed method, the pictures of the road scene taken by our team are used as the original images of the visualization results.

As shown in Fig. 13, compared with other segmentation methods, our segmentation method has a better segmentation performance for overlapping vehicles and incomplete road surfaces. Moreover, compared with BiseNet2, our method did not generate a significant amount of segmentation noise. This indicates that the proposed segmentation method can obtain high-level semantic information, which enhances the differences between categories.

Figure 13 Visualization results of semantic segmentation methods in complex road scenes.

(A) Is the original image; (B) is the segmentation image of the RTFormer; (C) is the segmentation image of the DeepLabV3+; (D) is the segmentation image of the SegFormer; (E) is the segmentation image of the FCN; (F) is the segmentation image of the PSPNet; (G) is the segmentation image of the BiseNet2; (H) is the segmentation image of the ICNet; (I) is the segmentation image of our segmentation method.

As shown in Fig. 14, the segmentation boundaries between different objects, such as the contours of pedestrians, vehicles, and road surfaces, were clearer in the segmentation image obtained using our method. Furthermore, our FSSFormer can correctly segment small, distant objects in an image. However, most segmentation methods fail to achieve this goal. Moreover, our segmentation method maintains good segmentation performance in both complex and simple scenes.

Figure 14 Visualization results of semantic segmentation methods in simple road scenes.

(A) Is the original image; (B) is the segmentation image of the RTFormer; (C) is the segmentation image of the DeepLabV3+; (D) is the segmentation image of the SegFormer; (E) is the segmentation image of the FCN; (F) is the segmentation image of the PSPNet; (G) is the segmentation image of the BiseNet2; (H) is the segmentation image of the ICNet; (I) is the segmentation image of our segmentation method.

Conclusion

To address the problem that road surface segmentation performance decreases in complex road scenes, we propose frequency-based semantic segmentation with a transformer (FSSFormer). First, we propose WSFA to enhance the performance of overlapping or incomplete target segmentation. Second, a cross-attention method combining spatial and frequency features was used to obtain boundary information. Finally, a parallel-gated feedforward network segmentation method is adopted to improve the accuracy of road surface segmentation in complex scenes. Extensive experiments demonstrated that our method improves mIoU, Precision and Recall by 2 %, 0.9 %, 3.1 % respectively compared with transformer-based method on the Cityscapes dataset. In addition, compared with other segmentation methods, FSSFormer has the better generalization performance which can be applied to the road surface segmentation under complex road conditions or recognition of different driving scenarios.

Road segmentation technology for unmanned driving has always been a key and challenging problem in computer vision tasks. Therefore, in the future, we will further improve the speed of our method. Specifically, we will examine whether the combination of convolution and attention can be used to replace the WSFA to speed up the operation.

Supplemental Information

Supplemental Information 1 FSSFormer model (training and testing code).

Supplemental Information 2 Photos taken by our team and the visualized results.

I would like to extend my sincere gratitude to my supervisor, Xiaomin Liu for her instructive advice and useful suggestions on my thesis. I am deeply grateful of her help in the completion of this thesis. High tribute shall be paid to Huaqi Zhao, whose profound knowledge of Computer science and Technology triggers my love for this subject and whose earnest attitude tells me how to learn it. I am also deeply indebted to Jeng-Shyang Pan for recommended journal.

Additional Information and Declarations

Competing Interests

Author Contributions

Data Availability

Rui Wang is employed by Dongfeng District People’s Court.

Huaqi Zhao conceived and designed the experiments, performed the experiments, analyzed the data, authored or reviewed drafts of the article, and approved the final draft.

Su Wang conceived and designed the experiments, performed the experiments, analyzed the data, performed the computation work, prepared figures and/or tables, authored or reviewed drafts of the article, and approved the final draft.

Xiang Peng conceived and designed the experiments, performed the experiments, analyzed the data, authored or reviewed drafts of the article, and approved the final draft.

Jeng-Shyang Pan conceived and designed the experiments, authored or reviewed drafts of the article, and approved the final draft.

Rui Wang conceived and designed the experiments, authored or reviewed drafts of the article, and approved the final draft.

Xiaomin Liu conceived and designed the experiments, performed the experiments, analyzed the data, authored or reviewed drafts of the article, and approved the final draft.

The following information was supplied regarding data availability:

The code is available in the Supplemental Files.

The Cityscapes dataset is available at Cityscapes and requires registration at their site to obtain access: https://www.cityscapes-dataset.com/register.

The COCO-Stuff dataset is available at GitHub:

https://github.com/nightrome/cocostuff.

The DarkZurich dataset is available at Trace Zurich:

https://www.trace.ethz.ch/publications/2019/GCMA_UIoU.

The ACDC dataset is available at ACDC:

https://acdc.vision.ee.ethz.ch/.

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
