# Peer review of "Road surface semantic segmentation for autonomous driving"

_PeerJ Computer Science, doi:10.7717/peerj-cs.2250_

## Round 0.1 · original submission · Minor Revisions

Dear authors,

Some concerns need to be addressed.

You are advised to critically respond to the comments point by point when preparing a new version of the manuscript and while preparing for the rebuttal letter.

Kind regards,
PCoelho

Reviewer 1 ·

Basic reporting

1. The first paragraph contains "semantic segmentation" at the beginning of nearly every sentence; it is recommended to simplify this.

2. In line 39, please add relevant references for "threshold selection" and "superpixel algorithms."

3. In line 167, the "SoftMax () functions" seems to be missing content within the parentheses.

4. Line 228 mentions "two publicly available datasets," but lists four datasets.

Experimental design

This paper provides a detailed experimental setting which includes the corresponding number of parameters, and analyzing the results.

Validity of the findings

In order to address the problem that road surface segmentation performance decreases in complex road scenes, this paper proposes some useful methods as follows:
1. proposes a frequency-based semantic segmentation with a transformer (FSSFormer);
2. propose a weight-sharing factorized attention to select important frequency features;
3. used a cross-attention method combining spatial and frequency features to obtain further detailed pixel information;
4. adopted a parallel-gated feedforward network segmentation method to encode the position information.

Cite this review as

Reviewer 2 ·

Basic reporting

3. To highlight the significance of your research in handling diverse real-world scenarios, please provide more detail on what constitutes 'complex scenes' in lines 35-37. Including illustrative examples or visual comparisons of different levels of scene, complexity would effectively demonstrate the adaptability and effectiveness of your method. Please clearly define what is complex scenes.
4. To improve clarity and provide readers with a more comprehensive understanding of your methodology, we suggest elaborating on the role of frequency features in the introduction and related work. You could include:
a. A brief overview of frequency features and their applications in similar tasks.
b. Discuss why you chose frequency features for this specific problem.
5. The reference style in lines 59-66 is inconsistent, with each bullet point referencing existing methods in a different way. This could confuse readers about which method you are referring to. Please revise this section to ensure a consistent and clear reference style throughout. In addition, there are already some other authors’ methods to overcome the above problems.
6. Before delving into the details of the proposed method, please expand lines 67-68 to include a brief overview of the experiment and the key metrics you aim to improve. This will give readers essential context and a clearer understanding of your research goals.
7. In line 164, please revise "For the high-" to either "For high-frequency and low-frequency features" or "For high and low frequency features" to ensure clarity and specificity.
8. Please expand on the statement in line 162, "Moreover, the segmentation noise was suppressed by this method," by providing supporting evidence or a more detailed explanation of how the method achieves this suppression.
9. While mathematical functions like “reshape(·)” and “concat(·)” are presented with parentheses in this document, the same format is not consistently applied to SoftMax. For consistency, please revise the text after line 167 to use the format “SoftMax(·)”.
10. Please revise the conclusion section to highlight the specific results (mIoU, Precision, Recall) that demonstrate the improvement achieved by your proposed method. Additionally, discuss this improvement's potential benefits or practical applications in relevant fields or scenarios.

Experimental design

1. To provide proper attribution and allow readers to access further information, please add a reference to the "Paddle1.8.0 framework" in the Experimental Setup section.
2. To enhance reader comprehension, please add a description of the red-dot data points to the captions of Figures 6, 7, and 8. This will ensure readers understand their significance within each figure's context.
3. For a more seamless reading experience, please consider placing Table 1 and Figure 9 together on the same page. This will allow readers to easily reference the data in the table while interpreting the visual information presented in the figure.
4. In Figure 9, the metrics' order (Recall, mIoU, Precision) differs from the consistent order used throughout the document (mIoU, Precision, Recall). Please align the chart's presentation with the established order to avoid confusion.
5. There appears to be a discrepancy regarding the hardware used in the experiments. Line 237 mentions that FPS is measured on a single RTX 2080Ti. However, Tables 2, 3, and 4 reference an RTX 2090Ti. Do Tables 2, 3, and 4 represent different evaluations or scenarios with RTX2090Ti?
6. To improve clarity and organization, Please rearrange the data in Tables 6 and 7. Start with models that are not directly comparable to your method due to differences in GPU, followed by those that are comparable (using the same GPU), and conclude with your proposed method in the last row.

Validity of the findings

No comments.

Cite this review as

---

## Round 0.2 · accepted · Accept

Dear authors, we are pleased to verify that you meet the reviewer's valuable feedback to improve your research.

Thank you for considering PeerJ Computer Science and submitting your work.

Reviewer 1 ·

Basic reporting

The authors have addressed the limitations of existing methods in complex traffic environments and have explained how this study fills that gap in the introduction.

Experimental design

The experimental setup is now detailed, including hardware configuration, software environment, and hyperparameter choices.

Validity of the findings

The authors have also revised the writing according to previous feedback and have added visualizations to aid reader comprehension.

Cite this review as

Reviewer 2 ·

Basic reporting

This is a revision paper. Authors have correct orignal errors. I suggest it is acceptable.

Experimental design

This is a revision paper. Authors have correct orignal errors. I suggest it is acceptable.

Validity of the findings

This is a revision paper. I suggest it is acceptable.

Cite this review as